# Non-Parametric Domain Adaptation Layer

## Abstract

Normalization methods spurred the development of increasingly deep and efficient archi-tectures as they reduce the distributions change during optimization, allowing for efficient training. However, most normalization methods cannot account for test-time distribution changes, increasing the vulnerability of the network concerning noise and input corruptions. As noise is ubiquitous and diverse in many applications, machine learning systems often fail drastically as they cannot cope with mismatches between training- and test-time activation distributions. The most common normalization method, batch normalization, is agnostic to changes in the input distribution during test time. This makes batch normalization prone to performance degradation whenever noise is present during test-time. Parametric correction schemes can only adjust for linear transformations of the activation distribution but not for changes in the distribution shape; this makes the network vulnerable to distribution changes that cannot be reflected in the normalization parameters. We propose an unsupervised non-parametric distribution correction layer that adapts the activation distribution and reduces the mismatch between the training and test-time distribution by minimizing the Wasserstein distance of each layer. We empirically show that the proposed method effectively improves the classification performance without the need for retraining or fine-tuning the model; on ImageNet-C it achieves up to 11 % improvement in Top-1 accuracy.

## 1 Introduction

Early on, Neural Networks (NNs) excelled at interpolating between training data points but failed at extrap-olating to regions beyond the training data (4; 13). Since sufficiently large datasets have not been available then, NN could only be successfully applied to tasks with well-known input distributions that would prevent any unpredictable behavior stemming from unknown data. The mismatch between training- and test-time distributions is known as *covariate shift*. More precisely, covariate shifts occur when a model $p(\mathbf{y}|\mathbf{x}, \theta)$ with parameters $\theta$, output $\mathbf{y}$, input $\mathbf{x}$, and a static conditional probability evaluates samples $\mathbf{x}$ from a different distribution than the training distribution (41; 43). The vulnerability of NNs to covariate shifts has been a major concern in the past.

In recent years, we have seen immense growth in the number of available input samples with the rise of big data and data augmentation techniques that have largely alleviated the mismatch between training- and test-time distributions (28). Covariate shifts remained a problem though, albeit in a different form: when training a neural network, every parameter update causes an internal distribution shift. This is particularly problematic for training Deep Neural Networks (DNNs) and often leads to convergence problems, as the internal shifts occur for every layer and are thus exacerbated by the large number of layers in DNNs. Thus, the main impact of the covariate shift moved from test-time to training-time.

In order to reduce internal covariate shifts, Batch Normalization (BN) aims to match the distribution of activations across different mini-batches (21). By doing so, BN greatly improved the convergence of DNNs and deep Convolutional Neural Networks (CNNs).[1] This, in turn, fueled the development of ever deeper and more capable architectures and – by reducing the dependence on the weight initialization – facilitated network training in an end-to-end fashion. Therefore, normalization became an elemental part of all deep

---

[1]Although later investigations of BN argue that the improved convergence properties actually result from smoothing the optimization surface (39) or from decoupling the length and direction of the gradients during optimization (26).

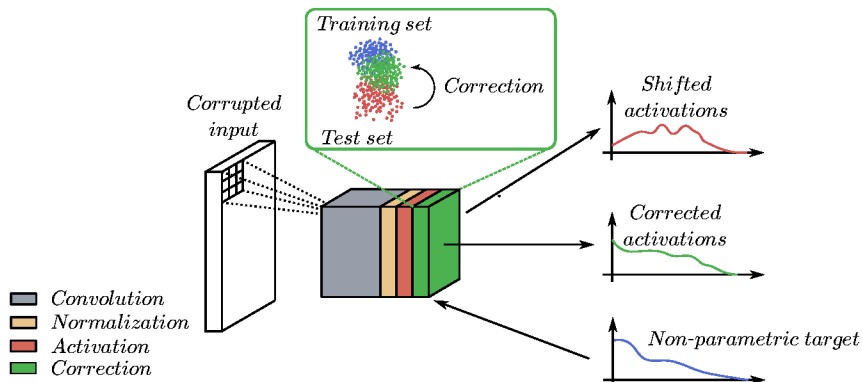

Figure 1: Illustration of the non-parametric correction layer (green). The correction transforms the shifted activations (red) towards the non-parametric target distribution (blue).

learning architectures. Increasing the depth of networks, however, increased the vulnerability to distribution shifts between training and test time again since BN estimates the normalization parameters (i.e., mean $\mu$ and variance $\sigma^2$) over all training samples and thus remains agnostic to changes in the input distribution during test-time. This explains why BN is inherently vulnerable to covariate shifts during test-time (e.g. caused by image corruptions (5)). Although one can improve robustness against specific image corruptions by data augmentation techniques, this concurrently reduces robustness against other corruption types (12; 5). Overall, this highlights the need for mitigating covariate shifts during test-time.

Some correction methods aim to move the test-time activations closer to the training-time activations by adapting the BN parameters to the input distributions (40). This, however, allows only for correcting linear transformations (e.g. mean shift or variance scaling) and not for arbitrary changes between distribution shapes. For discrete data such as grayscale or RGB images, histogram normalization methods can be used to transform and correct the distribution of an image (52; 25; 1). Note, that histogram normalization methods empirically construct a Probability Density Function (PDF) and are thus problematic to apply on continuous data (as the PDF depends largely on the chosen binning). Continuous data, as for example in the activation maps, therefore requires alternative correction methods that do not need access to the PDF. The Wasserstein distance provides such a method, as it only relies on samples drawn from the PDF and not on the distribution itself.

We propose a non-parametric distribution correction layer that utilizes the Wasserstein distance to reduce test-time distribution mismatches of an arbitrary form (see Figure 1).[2] Therefore, we consider an energy minimization scheme to find a maximum a-posteriori estimate, as used in image denoising (35; 6; 38; 51). This scheme balances two objectives and aims to approximate the target distribution well while retaining as much information as possible. The distribution mismatch of the activations is reflected in the Wasserstein distance between clean and corrupted samples (9; 2; 40). Since we do not have access to the clean samples in practice, we need to use a surrogate target distribution. Constructing a suitable target distribution that meets both objectives, however, is a challenging task. We choose to represent the typical activation distributions during training by their Wasserstein barycenters. To improve the correction, we collect multiple separate target distributions, one over the entire layer and one for each individual channel. Then we correct the test-time distributions by minimizing the one-dimensional Wasserstein distance to these target distributions. Note that we can compute this distance analytically by sorting the activations from both distributions.

Our proposed method corrects the distribution shape in an unsupervised fashion and without the need for retraining or fine-tuning of the models. This is fundamentally different from semi-supervised methods that retrain the model (either the normalization or the entire model parameters) based on test samples (31; 44; 48). Moreover, semi-supervised methods require multiple samples of the identical corruption type to achieve state-of-the-art performance; if only a few samples are available or if the samples contain random corruption types,

---

[2]Code will be made publicly available on acceptance.

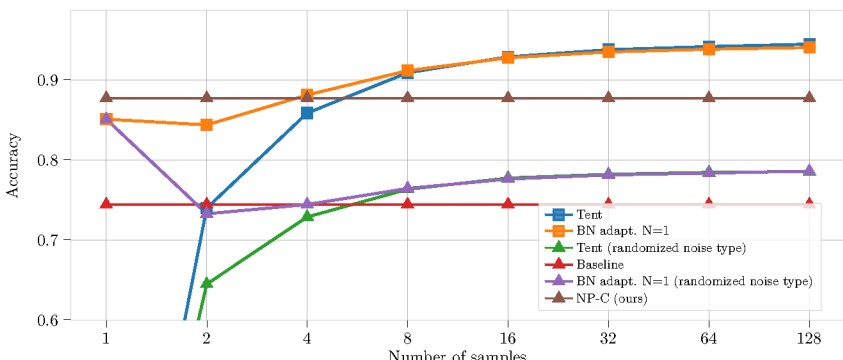

Figure 2: Comparison of the performance of three different domain adaptation methods on the corrupted MNIST dataset: Tent (48), BN adapt. (40), and our proposed non-parametric domain adaptation Layer (NP-C) for different numbers of used samples. The experiments marked by squared were conducted with all samples belonging to the same noise type, whereas the experiments marked by the triangles use the same dataset, but randomize the noise type.

semi-supervised correction methods often fail. That is often not visible in experiments, as the used datasets are sorted by corruption type and the separate types are evaluated successively. In contrast, our proposed method requires only a single *sample under test* to perform domain adaption and thus it does not suffer from similar problems. To further illustrate this, we compare the sample efficiency of different domain adaptation methods in Figure 2: experiments with identical corruption types for all samples are marked by squares; experiments with randomly shuffled corruption types are marked by triangles. The accuracy of semi-supervised methods suffers considerably when confronted with random corruption types and reduces by almost 16 % in the case of 128 samples (further experiments are presented in Section 4). There are various scenarios in which one cannot expect to see samples with identical (or slow-changing) corruption types. It would, for example, require a substantial computational overhead to sort and group images into different noise categories in server-based applications with multiple clients. This highlights the importance of domain adaptation methods that require information from only a single *sample under test*. We will thus focus particularly on this more general setting in this paper.

In our experiments, we show that our proposed method consistently improves robustness against input corruptions and outperforms parametric correction methods. We evaluate and compare our method on several corrupted versions of standard image classification datasets, MNIST-C, CIFAR-10-C, ImageNet-C. Furthermore, we exemplary analyze the proposed correction layer and provide insights into the underlying principles. To summarize our contributions:

- We propose an unsupervised non-parametric correction layer to mitigate distribution mismatches, caused by image corruptions, using only the information of the sample under test.

- We provide insights into the correction layer by evaluating activation distributions and activation maps with and without the correction. We further validate that the correction layer reduces the mismatch between corrupted and clean samples.

- In our experiments we analyze the impact of image corruptions in different CNN architectures using the corrupted datasets MNIST-C,CIFAR-10-C and ImageNet-C.

## 2   Related work

While today's deep learning methods achieve state-of-the-art performance for most machine learning benchmarks they lack generalization capabilities to different data domains and are vulnerable against corruptions and noise (10; 16; 33; 22). As robustness is critical in many real-world applications, various methods have

been developed to correct for the effects of domain shifts. These correction methods can be grouped into three main categories: data augmentation methods, learning-based methods, and normalization methods that aim to correct the statistics within the network.

**Data augmentation** improves robustness by synthetically increasing the coverage of the input space. In the simplest case, training samples are augmented – by the application of affine transformations or expected noise types – and explicitly included into the training set (3; 42; 28). More elaborate methods optimize the augmentation or use an additional neural network to increase the degree of realism of the augmented images (8; 17; 15). The application of data augmentation, however, is somewhat limited as improving robustness against one corruption type can decrease robustness against others (12; 5).

**Learning-based approaches** for improved robustness and prediction stability, comprise a large class of methods. Many are based on representation learning techniques, which aim to minimize the difference between source and target representations (47; 32; 45; 23; 24; 50; 37). Another line of research utilizes unlabeled data from the target domain to bridge the gap between domains by unsupervised and adversarial training (10; 11; 46; 7; 18; 19; 45). Others use semi-supervised methods and retrain the model based on test samples to reduce domain shifts caused by image corruptions (31; 44; 48). While these methods can be very effective in mitigating domain shifts, they require multiple unlabeled samples from the test set during retraining.

**Adapting normalization parameters** based on test-time statistics has recently attracted increasing interest (30). Adjusting the normalization parameters in an unsupervised fashion can drastically improve the model robustness (particularly for batch normalization). The influence of input corruptions on networks using batch normalization has been investigated in (5); this further led to a domain adaption method for the normalization that improves the robustness of CNNs against corruptions. This adaption can substantially improve the network's performance but requires a re-computation of the batch normalization parameters for each domain adaption. A similar approach was taken by (40), which adapts the normalization parameters based on test-time statistics and can achieve improved robustness using only a single test sample. The distribution correction is, however, limited to a shift and scaling.

## 3 Distribution correction for Convolutional Neural Networks (CNNs)

Input corruptions effectively introduce a distribution mismatch between training and test set. Thus, to make models more robust against input corruptions, one can try to reduce this distribution mismatch. Although such distribution corrections can indeed improve the classification performance of CNN architectures, most existing approaches struggle to do this effectively, since they are restricted to parametric distributions (e.g. Gaussians). Consequently, these methods can only correct for mismatches reflected in the distribution parameters (e.g., the mean and variance).

In order to mitigate more general distribution mismatches of the test-time activations $\mathbf{a}$, we must find an effective way to suppress noisy activations while maintaining the classification performance in the subsequent layers. Therefore, we will formulate this problem as a probabilistic denoising problem. We first need to approximate the a-posteriori distribution,

$$p(\tilde{\mathbf{a}}|\mathbf{a}) = \frac{p(\mathbf{a}|\tilde{\mathbf{a}})p(\tilde{\mathbf{a}})}{p(\mathbf{a})}, \tag{1}$$

of the corrected activations $\tilde{\mathbf{a}}$ given the activations $\mathbf{a}$. Then, we can use the maximum a-posteriori estimate to determine the corrected activations; this is a well-established technique in image denoising (35; 6; 38; 51). For our considerations, we recast the maximum a-posteriori problem into an equivalent energy minimization problem to simplify the optimization procedure. We assume that the prior, likelihood, and posterior come from an exponential family using a Gibbs measure so that

$$p(\tilde{\mathbf{a}}) = e^{-\frac{\mathcal{R}(\tilde{\mathbf{a}})}{T}}, \ p(\mathbf{a}|\tilde{\mathbf{a}}) = e^{-\frac{\mathcal{D}(\mathbf{a}|\tilde{\mathbf{a}})}{T}}, \ p(\tilde{\mathbf{a}}|\mathbf{a}) = e^{-\frac{E(\tilde{\mathbf{a}}|\mathbf{a})}{T}}. \tag{2}$$

Note that we can omit the evidence term $p(\mathbf{a})$, as we do not perform model comparison. Then, by applying the logarithm to Eqn. 2 and by multiplying all terms with $-\frac{1}{T}$, we arrive at an energy minimization problem

$$\tilde{\mathbf{a}}^* = \arg\min_{\tilde{\mathbf{a}}} E(\tilde{\mathbf{a}}|\mathbf{a}), \tag{3}$$

with the optimal activation map $\tilde{\mathbf{a}}^*$ at its minimum. The energy $E(\tilde{\mathbf{a}}|\mathbf{a})$ is composed of two terms $\mathcal{R}(\tilde{\mathbf{a}})$ (corresponding to the prior) and $D(\mathbf{a}|\tilde{\mathbf{a}})$ (corresponding to the likelihood) so that

$$E(\tilde{\mathbf{a}}|\mathbf{a}) = \mathcal{D}(\mathbf{a}|\tilde{\mathbf{a}}) + \mathcal{R}(\tilde{\mathbf{a}}). \tag{4}$$

For this form, we must, on the one hand, specify a suitable prior term $\mathcal{R}(\tilde{\mathbf{a}})$ that reduces the covariate shift in each layer without restricting the network (see Section 3.1). The data likelihood term $\mathcal{D}(\mathbf{a}|\tilde{\mathbf{a}})$, on the other hand preserves the spatial correlations of the activation maps and prevents the loss of valuable information (see Section 3.3). By minimizing both terms jointly, one can achieve an optimal trade-off between minimizing the covariate shift and retaining the underlying information.

### 3.1 Constructing a non-parametric prior term

Typically, the activation distributions in CNNs cannot be represented well by parametric distributions. Consequently, any correction method relying on a parametric approximation $q_\theta(\mathbf{t})$ with target values $\mathbf{t}$ will not capture the shape of the activations prior $p(\tilde{\mathbf{a}})$. Subsequent layers are thus exposed to different input distributions than during training and suffer from the corresponding covariate shift. Ideally the target distribution $q(\mathbf{t})$ should enforce similar (corrected) distributions as during training since any mismatch might outweigh the benefits of the noise-reduction otherwise. If $q(\mathbf{t})$ should resemble the non-parametric distribution from the training set, however, it must be non-parametric as well.

The Wasserstein distance proves to be particularly well-suited for a novel correction method: not only does it allow to effectively minimize the mismatch between the distributions during training- and test-time, but it also provides an elegant way of representing a non-parametric distribution in one dimension. Although we would ideally have access to the distributions of the clean image, this is not possible in practice. Instead we need to construct surrogate target distributions. As the Wasserstein distance can only be analytically represented in one dimension, we need to find representative slices through the high dimensional activation distributions, where each slice serves as a surrogate target distribution.

In order to obtain an appropriate surrogate target distribution from our training set, we consider two statistics: first, the layer-wise statistics where the activations are flattened to create a single distribution across the height $H$, width $W$, and channel $C$ dimension of the layer, and second channel-wise statistics that contain $H \times W$ values for each of the channel statistics, resulting in $C + 1$ target distributions per layer. To create useful target distributions, we need to ensure that the distributions only differ in shape but not in their mean. This is especially important when considering channel-wise statistics, as – depending on the input features – the individual means can vary significantly. Therefore, we remove the mean values before collecting the $N$ activations, i.e.,

$$\mathbf{a}'^{(m)} = \mathbf{a}^{(m)} - \frac{1}{N}\sum_{i=1}^{N} a_i^{(m)}, \tag{5}$$

where $N$ corresponds to the $N_l = H \times W \times C$ values for the layer-wise statistic and to the $N_c = H \times W$ values for the channel-wise statistics.

To find a well-suited prior distribution, we must first sort the $N$ activations $\mathbf{a}'$ for each training sample $m$ in ascending order. Let $[\cdot]$ be the vector of all corresponding elements, then

$$[a_{(i)}'^{(m)}] = \mathrm{sort}(\mathbf{a}'^{(m)}), \tag{6}$$

where $a_{(i)}^{(m)}$ are the sorted activations with $a_{(i)} < a_{(i+1)}$. Note that the subscript $(i)$ always denotes sorted values.

Next, we utilize the Wasserstein barycenter, i.e., the distribution that minimizes the sum of the Wasserstein distances $W$ over all $M$ training distributions (9; 2):

$$\min_q \sum_{m=1}^{M} W_d\big(q(\mathbf{t}'), p(\mathbf{a}'^{(m)})\big), \tag{7}$$

to construct the target values $t_{(i)}$ for the correction of the *sample under test.* In one dimension, the Wasserstein barycenter is simply the average over the order statistics of each sample $m$ so that

$$t'_{(i)} = \frac{1}{M} \sum_{m=1}^{M} a'^{(m)}_{(i)}. \tag{8}$$

Finally, to account for the mean correction of $t_{(i)}$ (see Eqn. 3.1), we must now re-add the channel mean $\mu^c$ and the sample specific mean $\mu^l$ to the respective target values,

$$t_{(i)} = t'_{(i)} + \mu. \tag{9}$$

This means that we collect separate mean values for all of the $C+1$ targets over the training set. The values $\mathbf{t} = [t_{(i)}]$ now represent our non-parametric target distributions $q(\mathbf{t^l})$ and $q(\mathbf{t^c})$ for the layer and channel-wise statistics respectively.[3]

## 3.2 Minimizing the Wasserstein distance

The one-dimensional Wasserstein distance between the target distribution $q(\mathbf{t})$ and the test-time distribution $p(\mathbf{a}^{(m)})$ of sample $m$ is given according to

$$W_d\big(p(\mathbf{a}^{(m)}), q(\mathbf{t})\big) = \left( \sum_{i=1}^{N} ||a^{(m)}_{(i)} - t_{(i)}||^r \right)^{\frac{1}{r}}, \tag{10}$$

where $t_{(i)}$ are the sorted target values and $a^{(m)}_{(i)}$ are the sorted test-time activations:

$$[a^{(m)}_{(i)}], \mathbf{j} = \text{sort}(\mathbf{a}^{(m)}), \tag{11}$$

and $\mathbf{j}$ are the indices of the activations that are required for mapping the updates from the sorted to the unsorted activations.

For $r = 1$; the Wasserstein distance between $p(\mathbf{a})$ and $q(\mathbf{t})$ in Eqn. 10 is minimized by,

$$\mathbf{\Delta}^{(m)} = [t_{(i)}] - [a^{(m)}_{(i)}]. \tag{12}$$

As the layer-wise $\mathbf{\Delta}^l$ as well as the channel-wise correction $\mathbf{\Delta}^c$ can create updates to the current activation map $\mathbf{a}$, we need to add both contributions:

$$\mathbf{\Delta}^{(m)} = \mathbf{\Delta}^{l,(m)} + \mathbf{\Delta}^{c,(m)}. \tag{13}$$

We now assign the updates $\mathbf{\Delta}^{(m)}$ to the corresponding unsorted activations using the mapping indices $\mathbf{j}$. To decrease the computational effort, we apply the correction after the ReLU activation function; thus, many activations are zero and sorting can be done more efficiently.

## 3.3 Data likelihood

Minimizing only the prior term might have undesired side effects and destroy important structures in the channels of the network, i.e., the spatial correlations. Therefore, the energy minimization needs to find a

---

[3]Whenever there is no ambiguity about the particular statistics, we will omit the superscript and simply write $q(\mathbf{t})$.

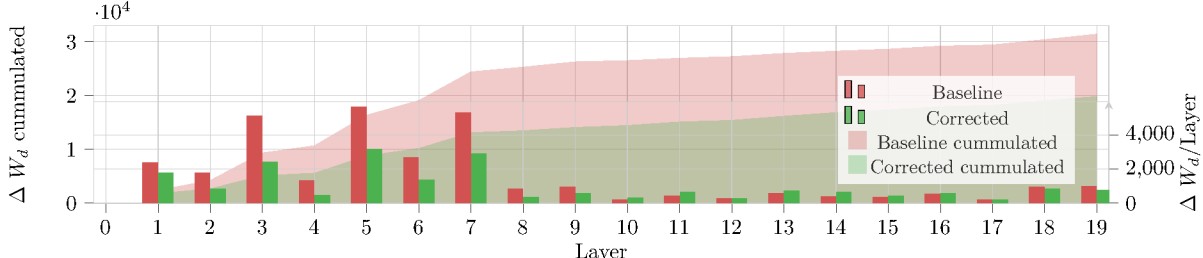

Figure 3: Difference in Wasserstein distance $\Delta W_d$ with $r = 1$ (as defined in Eqn. 10) between the activation distributions of clean images $\mathbf{t}$ and noisy images $\mathbf{a}$ (with Gaussian noise) for a ResNet20 trained on CIFAR-10. We compare the layer-wise and the cumulative distance with (green) and without (red) our correction.

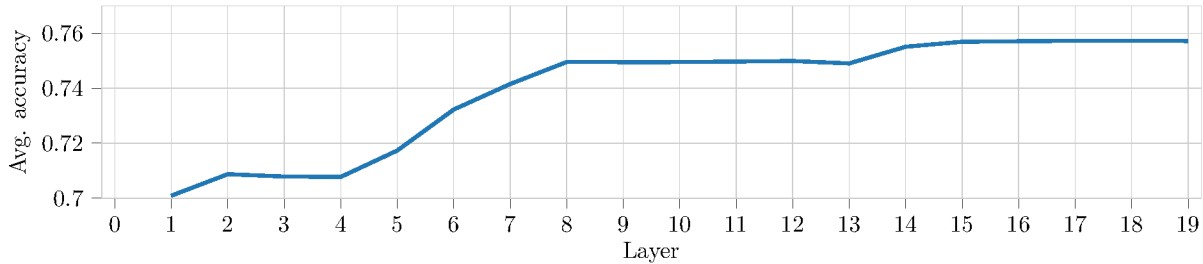

Figure 4: Average classification performance of a ResNet20 trained on CIFAR-10, for the corrupted CIFAR-10 dataset, as a function of the number of used NP-C layers. Correction layers are added successively from the first to the last layer.

trade-off between matching the distributions and conserving the spatial correlations. We achieve this by considering a likelihood term $\mathcal{D}(\mathbf{a}|\tilde{\mathbf{a}})$, modeled by

$$\mathcal{D}(\mathbf{a}|\tilde{\mathbf{a}}) = \frac{1}{2}||\mathbf{a} - \tilde{\mathbf{a}}||^2 \tag{14}$$

that conserves the structure in the data. This expression is also straightforward to minimize by exploiting the gradient

$$\frac{d\mathcal{D}}{d\mathbf{a}} = \mathbf{a} - \tilde{\mathbf{a}}. \tag{15}$$

### 3.4 Correction algorithm

The correction algorithm minimizes the energy,

$$E(\tilde{\mathbf{a}}|\mathbf{a}) = \lambda_1 \cdot \mathcal{D}(\mathbf{a}|\tilde{\mathbf{a}}) + \lambda_2 \cdot \mathcal{R}(\tilde{\mathbf{a}}), \tag{16}$$

where we choose $\lambda_1$ for the weighting factor of the prior-update, and $\lambda_2$ for the likelihood-update. These weighting factors determine the relative importance of the corresponding terms. In practice, the weighting factors should be chosen to achieve good classification performance, balancing the correction and the information loss in the activation maps. The proposed algorithm computes the corrections and successively minimizes the mismatch between the distributions $p(\mathbf{a}^{(m)})$ and $q(\mathbf{t}^1)$, and between $p(\mathbf{a}^{(m)})$ and $q(\mathbf{t^c})$ in a layer-wise fashion. That is – starting from the input layer – we correct the activations by performing the updates. We perform only a single update for both the prior and the likelihood terms in each layer. This is sufficient, as the layered nature of CNNs unrolls the correction over the entire architecture. As the correction is applied at test-time in an unsupervised fashion, it can easily be retrofitted into existing models by adding the correction layer and collecting the target distributions $q(\mathbf{t}^1)$ and $q(\mathbf{t^c})$ for each layer, using the training set. We present pseudocode of the algorithm below:

---

**Algorithm 1** Activation Correction Algorithm

---

**Input:** Activations $\mathbf{a}$, sorted target value vectors $\mathbf{t^l}$ and $\mathbf{t^c}$, step-sizes $\lambda_1, \lambda_2$, number of channels $C$
**Output:** Corrected activations $\tilde{\mathbf{a}}$

$\boldsymbol{\Delta^l} \leftarrow \textsc{Calculate correction}(\mathbf{a}, \mathbf{t^l})$
**for** $k < C$ **do**
    $\boldsymbol{\Delta^c_k} \leftarrow \textsc{Calculate correction}(\mathbf{a}_k, \mathbf{t^c_k})$
**end for**
$\boldsymbol{\Delta} \leftarrow \boldsymbol{\Delta^l} + \boldsymbol{\Delta^c}$
$\mathbf{a'} \leftarrow \lambda_1 \boldsymbol{\Delta}$
$\tilde{\mathbf{a}} \leftarrow \mathbf{a'} + \lambda_2(\mathbf{a} - \mathbf{a'})$

**function** $\textsc{Calculate correction}(\mathbf{a}, \mathbf{t})$
    $N \leftarrow \text{len}(\mathbf{a})$
    $[a_{(i)}], \mathbf{j} \leftarrow \text{sort}(\mathbf{a})$
    **for** $i < N$ **do**
        $k \leftarrow j_i$
        $\Delta_k \leftarrow (t^l_{(i)} - a_{(i)})$
    **end for**
    **return** $\Delta$
**end function**

---

## 4 Experiments

In our experiments, we first investigate our proposed method concerning its distribution matching capabilities by empirically analysing how the correction affects the activation maps (see Section 4.1). Second, we consider models trained on MNIST and CIFAR-10 and evaluate how well and how consistent our correction performs on the corrupted variants of these datasets (27; 29) (see Section 4.2; we further analyze the impact of different noise types.[4] Third, we consider different CNN architectures (pre-trained on ImageNet and with weights provided by Keras) and evaluate the performance of our proposed correction layer on the corrupted ImageNet (ImageNet-C) dataset (see Section 4.3). Finally, we discuss the current limitations of our proposed method (see Section 4.4). All datasets are publicly available on TensorFlow datasets (34; 16).[5]

### 4.1 Analyzing the correction on activation maps

The goal of our approach is to reduce the distribution shift within the network; to verify this, we analyze the activation maps and their distributions before and after applying the correction. First, we verify if our proposed surrogate distributions are expressive enough to decrease the Wasserstein distance to the clean activation distributions. Therefore, we compare the activations of a ResNet-20 trained on CIFAR-10 (see experimental detail in Section 4.2) from clean images with those of noisy images containing Gaussian noise. In Figure 3, we see that the correction successfully minimizes the Wasserstein distance and brings the activations from the corrupted samples closer to the clean ones. This is especially pronounced within the first seven layers, for which the correction reduces the distance by almost one-half. Layers 8-17 only seem to result in minor changes in Wasserstein distance. However, if we analyze the average classification accuracy with respect to the number of used NP-C layers in Figure 4, we see that with the exception of layer 13, performance still improves or is stable within those layers. This indicates that a small Wasserstein distance does not directly translate to good classification performance, which might be an effect of only using slices through the high dimensional activation space.

Next, we analyze the effect of the correction on the activation maps and the underlying distributions. In Figure 5, we see an example image passed through a ResNet-50 with and without a correction layer. This

---

[4]Models were trained on an NVIDIA Tesla V100 GPU.
[5]More detailed results are provided in the Appendix.

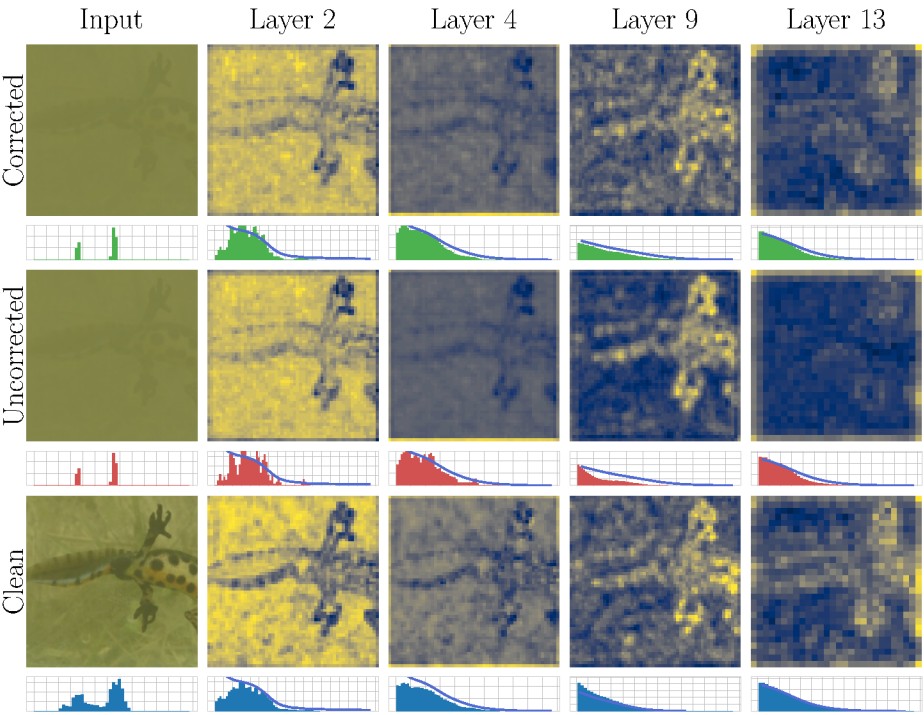

Figure 5: Comparison between the activation maps and histograms of four different layers with increasing depth of a ResNet-50 trained on ImageNet. For creating the activation map images we averaged over all channels. The input image (first row) is from the ImageNet-C dataset containing the highest (severity 5) input corruption regarding contrast, the difference in classification performance between corrected and uncorrected activations for this corruption type is 27.45 % Top-1 accuracy.

image includes a contrast corruption of severity 5 (highest), for which our correction layer is particularly effective, achieving 27.45 % better Top-1 accuracy. The distribution histograms show that the correction layer successfully increases the similarity between sample $p(\mathbf{a}^{(\mathbf{m})})$ and target $q(\mathbf{t})$ (blue curve), while – to an extent – also smoothing the shape of the distribution. Note that this is to be expected, as the correction towards the target distributions does not preserve the locality of the features. This does not affect the classification much, since the spatial structure is most relevant in the first layers, for which the correction has only little impact on the locality. Overall, the correction layer introduces richer features and thus helps to achieve good classification performance.

## 4.2 Corrupted MNIST and CIFAR-10 classification

In this section, we analyze the performance and the variability of randomly initialized models on the corrupted variants of the MNIST and CIFAR-10 dataset with and without our correction layer. We trained ten models using Batch Normalization (BN) and ten models using Group Normalization (GN) for both datasets (49). The randomly initialized models were trained on the clean versions of the respective datasets using the schemes presented in Sections 4.2.1 and 4.2.2. Furthermore, we compare the results of our non-parametric correction layer (NP-C) to another unsupervised correction method. This alternative correction method (see (40)) relies on a parametric BN adaptation (BN adapt.) and blends the training and test-time parameters to correct for distribution shifts and scaling. We compare both methods for the special case of $n = 1$; i.e., when only a single *sample under test* is available.

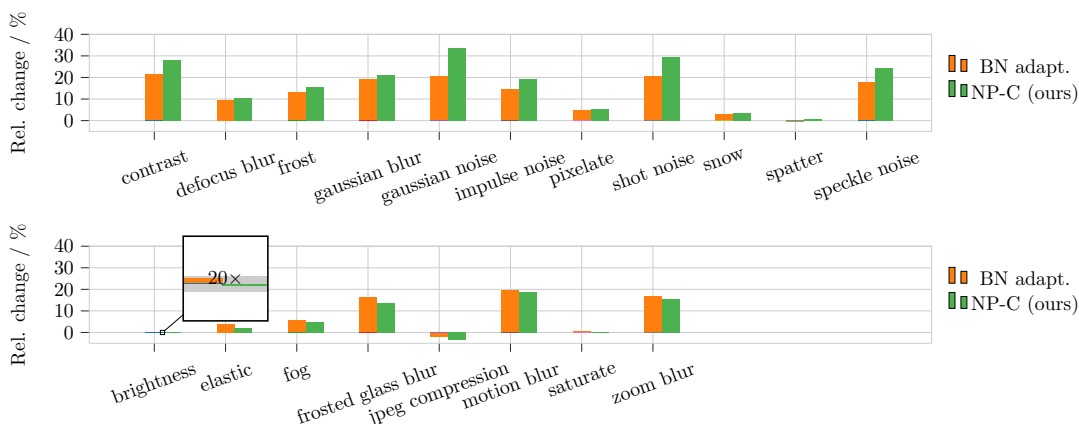

Figure 6: Relative change of the classification accuracy of a ResNet-20 trained on CIFAR-10 for specific corruption types. The top figure shows the corruption types for which the non-parametric correction (NP-C) performs better, and the bottom figure shows the corruptions where NP-C performs worse compared to the parametric BN adaptation method (40).

### 4.2.1 MNIST

We trained 10 randomly initialized ResNet-20 models for 50 epochs using an SGD optimizer with a base learning rate of 0.1 on the clean MNIST dataset (29). We decayed the learning rate after 25 and 40 epochs by a factor of 0.1. The input data was normalized to a range $[0, 1]$. The corrupted MNIST dataset contains 15 different corruption variants of the original MNIST images (see (34) for details). It contains 10.000 grayscale images of size $28 \times 28$ per corruption; we used all of them for our evaluations. For this experiment, we choose different weighting parameters $\lambda_1$ and $\lambda_2$ for each normalization method. We performed a coarse hyperparameter search fixing the reconstruction weight at $\lambda_2 = 0.8$ for all experiments. For BN we found $\lambda_1 = 1.2$ optimal, while for GN $\lambda_1 = 1.0$ provided the best results. We also optimized the parametric BN adaptation concerning its hyperparameter $N$, representing the *pseudo sample size* for samples from the training set. Here $N = 1$ resulted in the best performance. For GN the parametric adaption is not possible as it is sample-based by default.

Table 1: Avg. classification accuracy of ResNets for the corrupted MNIST dataset; we compare our method (NP-C) to the method presented in (40) (BN adapt.) and to the baseline (BN and GN).

|  | Accuracy |
|---|---|
| BN | 75.42±0.82 |
| BN adapt. | 88.22±1.35 |
| BN NP-C (ours) | 89.09±0.87 |
| GN | 92.22±0.96 |
| GN NP-C (ours) | **92.66±1.03** |

The average classification results are presented in Table 1, detailed results are shown in Appendix A.1. We see that both the parametric (BN adapt.) and the proposed non-parametric correction (NP-C) substantially improve the performance compared to networks using BN. Here the results of our method outperform the parametric BN adaptation by about 0.9 % on average and show less variability across all models. We further observe that networks using GN are inherently more robust, as they can automatically adjust to changing means and variances of the underlying activation distributions. Nonetheless, our correction method is capable of improving the performance even further. This indicates the adverse influence of non-parametric distortions on classification performance.

Table 2: Classification accuracy of ResNets for the corrupted CIFAR-10 dataset for different severity levels; we compare our method (NP-C) to the method presented in (40) (BN adapt.) and to the baseline (BN and GN).

| Method | Level 1 | Level 2 | Level 3 | Level 4 | Level 5 | Avg. |
|---|---|---|---|---|---|---|
| BN | 83.78±0.62 | 77.38±0.65 | 71.06±0.81 | 63.61±1.01 | 52.07±1.11 | 69.7±0.88 |
| BN adapt. | **84.42±0.35** | 80.39±0.42 | 76.57±0.54 | 71.83±0.67 | 63.25±0.86 | 75.35±0.6 |
| BN NP-C (ours) | 84.07±0.39 | **80.43±0.44** | 77.1±0.55 | 72.57±0.67 | 65.31±0.88 | 75.94±0.62 |
| GN | 83.33±1.07 | 79.74±1.2 | 76.77±1.28 | 72.54±1.31 | 65.83±1.38 | 75.64±1.25 |
| GN NP-C (ours) | 83.05±1.01 | 79.86±1.09 | **77.26±1.13** | **73.51±1.12** | **67.41±1.17** | **76.88±0.95** |

### 4.2.2 CIFAR-10

We trained 10 randomly initialized ResNet-20 models for 300 epochs using an SGD optimizer with a base learning rate of 0.1 on the clean CIFAR-10 dataset (27). During training, we decayed the learning rate after 150 and 225 epochs by a factor of 0.1. The input data was normalized to zero mean and unit variance and a widely used standard data augmentation scheme was performed (14; 20). The corrupted CIFAR-10 dataset contains 19 different corruption variants (see (16) for details). Additionally, the corrupted CIFAR-10 dataset features five different levels of corruption severity for each corruption type. For every corruption type and severity, it contains 10000 RGB images of size 32×32. In Figure 7, we see the evaluation results of

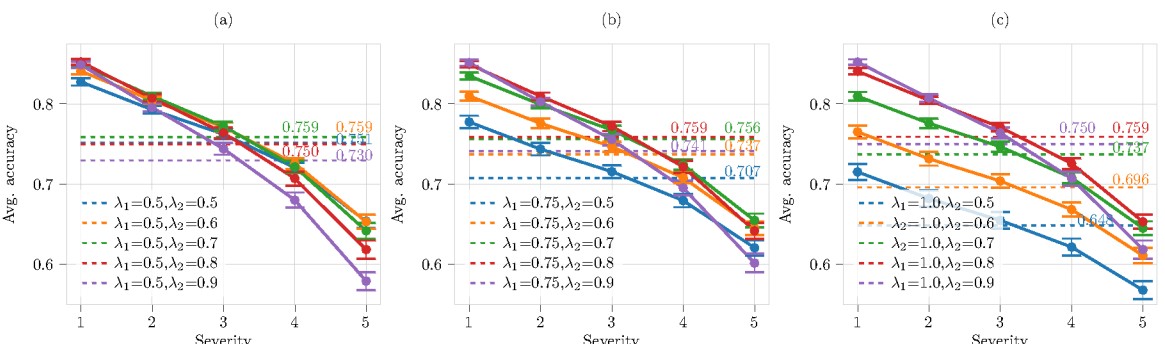

Figure 7: Comparison of the evaluation of the CIFAR-10 dataset for different combinations of $\lambda_1$ and $\lambda_2$ for ResNet-20 models trained on CIFAR-10. (a) uses $\lambda_1 = 0.5$, (b) uses $\lambda_2 = 0.75$, and (c) uses $\lambda_1 = 1.0$. Dashed lines indicated average performance over all corruption types and models.

different combinations of $\lambda_1$ and $\lambda_2$ for a ResNet-20 using BN. We see that multiple combinations $\lambda$ values reach similar results, peaking at about $\approx 76\%$ accuracy averaged over all corruption types and models. This indicates that coarse fine-tuning is sufficient as the optimum can be found quite easily. We selected $\lambda_1 = 1.0$ for the BN, and $\lambda_1 = 0.7$ for the GN models and $\lambda_2 = 0.8$ for both cases. In the following evaluations, we again optimized the parametric BN adaptation concerning its hyperparameter $N$, representing the *pseudo sample size* for samples from the training set. Here $N = 4$ resulted in the best performance.

The results are presented in Table 2, and detailed results in Appendix A.2. We see that our correction layer had the most improvement for the GN models, outperforming the baseline method by over 1% accuracy. This is particularly interesting, as CIFAR-10 is more complex than MNIST; this might indicate that there is a close connection between the complexity of the dataset and the amount of non-parametric distortions. Comparing the results of BN and GN, we further see that BN with our proposed method (BN NP-C) performs better than models simply using GN. This is different from MNIST (cf. Table 1) and might indicate that the trade-off between robustness and clean-data accuracy tilts towards clean-data performance for more complex data. Consequently, the performance of GN is hampered by its focus on robustness.

In Figure 6, we show the relative change of the average accuracy on specific corruption types. Here especially the classical noise types such as Gaussian noise, impulse noise, and shot noise, but also less stochastic

Table 3: Classification accuracy of a ResNet-50 for *ad hoc* domain adaptation with ImageNet-C; we compare our method (NP-C) to the method presented in (40) (BN adapt.) and to the baseline (BN).

| Architecture | Method | Top-1 | mCE |
|---|---|---|---|
| DenseNet-121 | BN | 45.11 | 70.16 |
| | BN adapt. | 46.46 | 68.49 |
| | NP-C (ours) | **46.89** | **67.95** |
| EfficientNet-B0 | BN | 38.88 | 77.87 |
| | BN adapt. | 44.83 | 70.44 |
| | NP-C (ours) | **49.88** | **64.04** |
| InceptionNet-V3 | BN | 50.33 | 63.44 |
| | BN adapt. | 53.03 | 60.08 |
| | NP-C (ours) | **53.24** | **59.80** |
| MobileNet-V3-L | BN | 37.64 | 79.44 |
| | BN adapt. | 38.42 | 78.77 |
| | NP-C (ours) | **40.10** | **76.48** |
| ResNet-50 | BN | 43.41 | 72.10 |
| | BN adapt. | 46.38 | 68.38 |
| | NP-C (ours) | **48.07** | **66.36** |

corruptions, such as contrast, benefit the most from our non-parametric correction. For the cases where our correction performed worse than BN adaptation (bottom), we see only a minor performance gap. Interestingly, both methods actually degrade the performance for the frosted glass blur corruption. This indicates that in this case, the performance degradation of the model is not or only to a small extent caused by distribution shifts.

## 4.3 Corrupted ImageNet classification

In this section, we analyze the effect of our proposed non-parametric correction on the performance of different CNN architectures (DenseNet-121, EfficientNet-B0, InceptionNet-V3, MobileNet-V3-L, ResNet-50) on the ImageNet dataset. We use pre-trained ImageNet weights, and pre-processing toolboxes provided by Keras Applications. We then add our correction layers after the activation layers. For the evaluation of the corrupted ImageNet (ImageNet-C) dataset, we again choose $\lambda_2 = 0.8$ and perform a coarse hyperparameter search using the ResNet50 architecture. Here $\lambda_1 = 0.2$ provided the best results. This indicates that $\lambda_1$ correlates with the complexity resulting from the image size of the used dataset. We again use the parametric BN adaptation method as our baseline, using a *pseudo sample size* of $N = 16$, as suggested in (40). We reuse this setting for all the investigated architectures. We adopt the mean corruption error (mCE) as an additional evaluation metric, as suggested by (16). This metric uses the AlexNet errors on the five severities of each corruption type $c$ on ImageNet-C, to adjust for the different difficulty levels of the corruption types for the classifier $f$ (28):

$$mCE_c = \frac{\sum_{i=1}^{5} E_{i,c}^{f}}{\sum_{i=1}^{5} E_{i,c}^{AlexNet}} \tag{17}$$

The ImageNet-C dataset contains 19 different corruption variants of the original dataset. It contains 50000 RGB images of size 224×224 per corruption that we used for our evaluations.

The results in Table 3 show, that our proposed correction layer can improve robustness for all of the investigated architectures, outperforming the parametric correction. While DenseNet-121 and InceptionNet-V3 perform almost on par with the parametric BN adaptation, the largest margin (11 % over baseline and ≈5 % over BN adaptation) was achieved for the EfficientNet-B0 architecture. This is particularly interesting, as the weighting factors were optimized for the ResNet-50 architecture. One possible explanation is that EfficientNet is the only architecture that uses a Swish activation function (36). As this function is not piece-wise linear, it also introduces more non-parametric distortions to the activation distribution.

Table 4: Evaluation times for different network architectures with and without the proposed correction layer.

| Correction applied | No | Yes |
|---|---|---|
| ResNet-20@CIFAR-10 | 0.2 ms | 3.7 ms |
| MobileNet-V3-L@ImageNet | 0.3 ms | 17 ms |
| EfficientNet-B0@ImageNet | 0.4 ms | 26 ms |
| ResNet-50@ImageNet | 0.5 ms | 38 ms |
| DenseNet-121@ImageNet | 0.6 ms | 67 ms |

### 4.4 Limitations and ethical concerns

The main limitation of this approach is the additional computational requirements. As the algorithm requires the sorting of $K$ activations, the complexity scales with $\mathcal{O}(K \log K)$. This causes an overhead, especially for datasets with large images, such as ImageNet. Table 4 shows the evaluation times for different architecture with and without the correction layer. For a ResNet-20 on CIFAR-10, the average evaluation time of a single sample increases from 0.2 ms to 3.7 ms on a single Tesla V100 GPU. This translates to approximately a factor of 19 for the evaluation time per image. On ImageNet the average evaluation time of the same system increases from a factor of around 57 for MobileNet-V3-L, to a factor of around 112 for DenseNet-121. This problem currently limits the method to relatively shallow networks but also opens a future research direction.

We do not have particular ethical concerns regarding our proposed method and do not expect a negative societal impact. One should note, however, that any method that makes image classifiers more robust inevitably broadens the field of potential unethical applications.

## 5 Conclusion and Outlook

We proposed a non-parametric domain adaptation layer based on the Wasserstein distance. It reduces the domain shift between test-time and training activations caused by input corruptions. The proposed method uses a maximum a-posteriori estimate, determined by minimizing the energy with respect to a data likelihood and a non-parametric prior term. Our proposed method works in an unsupervised setting for randomized corruption types and can be retrofitted into existing networks without retraining. In our experiments, we discussed the effects of distribution mismatches between clean and corrupted data and demonstrated how our correction layer successfully reduces this mismatch. On corrupted input (for MNIST-C, CIFAR-10-C, and ImageNet-C) the non-parametric correction layer consistently improved the classification performance and outperformed parametric approaches. In the future, we aim to consider alternative target distributions, improve the efficiency, and extend our method beyond convolutional neural networks.

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
