# OpenReview forum: "Non-Parametric Domain Adaptation Layer"
_TMLR — Rejected by TMLR_

### Review · Reviewer_Pbnu · 2022-04-18

**Summary Of Contributions:**

The authors propose a novel non-parametric adaptation layer for a neural network that can produce robustness to corruptions in the input image using only a test sample. The method involves predicting the maximum-a-posteriori (MAP) estimate of the corrected activations.The MAP estimate requires constructing a non-parametric prior and data likelihood term. The non-parametric prior requires computing the Wasserstein distance between the target distribution and the test-time distribution. The data-likelihood term requires a mean square error minimization estimate. The overall algorithm produces better recognition results on robustness corruption datasets compared to batch normalization and group normalization based methods.


**Requested Changes:**

- Comparison could be carried out with domain adaptation baselines like digit recognition, GTA to Cityscape as done in the TENT paper [1].

- One could consider a few SOTA denoising methods in [2] for fair comparison.

- Comparison should be done against also more recent test time adaptation methods like [3,4] and other papers which refer the TENT paper.

- TSNE plots could be shown to highlight the domain differences and adaptation.

- How would the method work if the network did not have batch normalisation layers as in [5] ?


References -

[1] Tent: Fully Test-Time Adaptation by Entropy Minimization., ICLR’21

[2] Brief review of image denoising techniques. Visual Computing for Industry, Biomedicine, and Art

[3] SITA: Single Image Test-time Adaptation

[4] Learning Instance-Specific Adaptation for Cross-Domain Segmentation

[5] High-Performance Large-Scale Image Recognition Without Normalization


**Strengths And Weaknesses:**

---- Strengths ----

- The method is simple and intuitive and can be added onto any network.
- The method is grounded on well-formulated statistics and optimization.
- The method produces improvement over the compared baselines.

---- Weaknesses ----

- The major weakness of this paper is that although this paper claims to be based on domain adaptation, it mainly focuses on noise adaptation and not the general domain adaptation benchmarks.

- Even if we consider the paper to be a noise adaptation paper, there has been no comparison with SOTA denoising methods which can be used to denoise an image and feed into the network.

- Comparison studies have been done only against Batch Normalization and Group Normalization.

- Additional ablation studies analysing the proposed method are missing. Please refer “Requested Changes” section for the details.

---

> ### Author Response · Authors · 2022-05-04
> **Response**
>
> Thank you for your review!
>
> General remarks:
>
> We agree with the reviewer's sentiment that the introduction needs to be improved. We will perform a major revision of the introduction and the related work section to improve readability, clarify the background, and better motivate our work. Moreover, we will improve Section 3 to clarify the details of our proposed approach.
> Additionally, we will perform additional benchmarks on a selected set of more realistic datasets that were suggested in the reviews, i.e CIFAR-10.1, ImageNet-V2, ImageNet-R, Cityscapes.
> After carefully reviewing the related works introduced by the reviewers [2][3][4], and comparing them to [1], we will also adjust our experimental setup on the ImageNet-C and CIFAR-10-C datasets, since these works use a 15 test + 4 holdout corruption setting, instead of a test/validation splitting for the 19 individual corruptions. Also, we will perform the CIFAR evaluations using a ResNet-26. These measures should improve comparability to other methods.
>
> Concerning your requested changes:
>
> - As mentioned above we will perform evaluations on the Cityscape dataset and additional datasets that contain more realistic corruptions.
>
> - As most denoising methods such as Wiener filter, Tikhonov denoising, or TV-norm denoising were designed to remove classical noise types such as Gaussian noise, impulse noise, and speckle noise, we expect them to perform well on these specific noise types. However, for other noise types such as spatter, or fog we do not expect them to improve results they might even degrade, also they will also introduce additional corruptions to the images which might again degrade model performance. We will provide an exemplary comparison for a selected denoising method.
>
> - Thank you very much for the references [2,5]; we were not aware of these works and will add them to the paper. They address the same issue, of single image adaptation, and underline the importance of this type of method. Also, they will serve as good baseline methods to compare against those that we will include in the paper.
>
> - TSNE plots for a selected model will be included
>
> - While most adaptation methods rely on modifying the BN parameters, our method does not. Therefore, it can be universally applied to any CNN architecture. This is a big advantage for architectures such as in [5]. Thank you again for pointing us towards this special use case for our method. We will mention this in the revised version of the paper.
>
> References:
>
> [1] Wang, D., Shelhamer, E., Liu, S., Olshausen, B. and Darrell, T., 2020. Tent: Fully test-time adaptation by entropy minimization. arXiv preprint arXiv:2006.10726.
> https://arxiv.org/pdf/2006.10726.pdf
>
> [2] Khurana, A., Paul, S., Rai, P., Biswas, S. and Aggarwal, G., 2021. SITA: Single Image Test-time Adaptation. arXiv preprint arXiv:2112.02355.
> https://arxiv.org/pdf/2112.02355.pdf
>
> [3] Zhang, M., Levine, S. and Finn, C., 2021. MEMO: Test Time Robustness via Adaptation and Augmentation. arXiv preprint arXiv:2110.09506.
> https://arxiv.org/pdf/2110.09506v2.pdf
>
> [4] Sun, Y., Wang, X., Liu, Z., Miller, J., Efros, A. and Hardt, M., 2020, November. Test-time training with self-supervision for generalization under distribution shifts. In International Conference on Machine Learning (pp. 9229-9248). PMLR.
> https://arxiv.org/abs/1909.13231
>
> [5] Zou, Y., Zhang, Z., Li, C.L., Zhang, H., Pfister, T. and Huang, J.B., 2022. Learning Instance-Specific Adaptation for Cross-Domain Segmentation. arXiv preprint arXiv:2203.16530.
> https://arxiv.org/pdf/2203.16530.pdf

---

### Review · Reviewer_bMwH · 2022-04-25

**Summary Of Contributions:**

The authors propose a non-parametric normalization method to alleviate the shift between the train and test distributions to improve model performance on corrupted images. For this, they calculate the mean activation per layer for the training data and seek to align the activation distribution of a corrupted sample with this target activation by reducing the Wasserstein distance between the two. They show results on corrupted versions of MNIST, CIFAR10 and ImageNet.

**Broader Impact Concerns:**

I do not have broader impact concerns for this publication.

**Requested Changes:**

I have written a detailed review in the above paragraph. In summary, I believe that the introduction, related work and the methods section need to be rewritten for clarity. I have found several wrong and unsupported statements, as well as misplaced or missed references in the related work section. The methods section (section 3) is hard to understand and should be made much more clear as it took me a lot of time to parse it. I am still not convinced I understood everything in section 3, so it definitely needs rewriting for clarity.

Considering the results, the gains on MNIST and CIFAR10 are below 1 p.p. which I consider to not be statistically significant, also given the large error margins. For CIFAR10, the currently SotA CIFAR-C numbers are around 93% without adaptation: https://robustbench.github.io/#div_cifar10_corruptions_heading. I argue that showing improvements on the order of below 1 pp for subpar networks is meaningless for the robustness community since it is not clear whether this method would generalize to better models. Given how small the improvements are, I do not expect major gains for the stronger models. The authors would need to show improvements for a SotA model, not the ResNet20 that they are using.

On ImageNet, the only model where their method seems to significantly improve over BN adapt is the EfficientNetB0 model, so overall their method does not seem to generalize to different architectures. Given how large the additional computational requirements are, especially on ImageNet, I currently consider the marginal improvements to not outweigh the costs. Especially since the authors claim that their method does not require model retraining, it is not clear whether there is actually any benefit at all, given how computationally intense their method is. The authors should evaluate their method on more models to see whether there are significant improvements on models besides the EfficientNetB0.

It seems that pages containing images are jpegs themselves (definitely not pdfs) as it is not possible to mark text which requires retyping paragraphs to cite them during the review. Please resubmit the paper with all pages in pdf format. I also cannot click the references, please fix this.

The authors did not state how they chose their hyperparameters. It reads as if they chose them on the test set. If this is the case, I consider the results to be meaningless. The authors should define a holdout validation set and choose their hyperparameters on it.

Overall, I would suggest the authors read my detailed comments where I stated my expectations on how this paper should be revised.

**Strengths And Weaknesses:**

It seems that pages containing images are jpegs themselves (definitely not pdfs) as it is not possible to mark text which requires retyping paragraphs to cite them during the review. Please resubmit the paper with all pages in pdf format. I also cannot click the references, please fix this.

### Comments to Introduction:
“Early on, Neural Networks (NNs) excelled at interpolating between training data points but failed at extrapolating to regions beyond the training data (4; 13).” It is wrong to state that “early on” the NNs had issues with extrapolation. This issue has not been solved yet and remains a problem.
„In recent years, we have seen immense growth in the number of available input samples with the rise of big data and data augmentation techniques that have largely alleviated the mismatch between training- and test-time distributions (28).” This statement is wrong. It is well known that models trained on ImageNet perform well on the ImageNet test set, but underperform on ImageNet-V2 (which was designed to not have a distribution shift), ObjectNet, ImageNet-A,-R,-C,-P,-D and are vulnerable to adversarial attacks. Common corruptions which are captured in e.g., ImageNet-C, are to be expected in real-world applications, and therefore, merely having more data (as in ImageNet) is not sufficient to alleviate the distribution mismatch between test and train.

“when training a neural network, every parameter update causes an internal distribution shift. This is particularly problematic for training Deep Neural Networks (DNNs) and often leads to convergence problems, as the internal shifts occur for every layer and are thus exacerbated by the large number of layers in DNNs. Thus, the main impact of the covariate shift moved from test-time to training-time. In order to reduce internal covariate shifts, Batch Normalization (BN) aims to match the distribution of activations across different mini-batches (21).” -> This is wrong. It has been shown that BN does not reduce internal covariate shift and works for different reasons, namely, by smoothing the loss landscape. Please see “How Does Batch Normalization Help Optimization?” by Santurkar et al. The footnote 1 does not do justice to Santurkar’s work as the internal covariate shift reduction by BN claim still stands in the foreground in the main text. I would suggest to reformulate the whole paragraph.

“Increasing the depth of networks increased their vulnerability to distribution shifts.” This needs citation and proof as I do not believe it to be true. In “Measuring Robustness to Natural Distribution Shifts in Image Classification” by Taori et al., the performance on the ImageNet validation set is strongly correlated with ImageNet-C accuracy (Fig. 5). As we know that larger models typically perform better on ImageNet, following the authors’ logic, we would expect worse relative performance for larger models with stronger ImageNet accuracy. We do not see this effect.

I have the impression that the authors mix the “internal covariate shift” occurring in course of training which BN was believed to reduce, and “covariate shift” occurring due to the mismatch of the training and test distributions which adaptive BN does alleviate. On page 1, the authors state that “In recent years, we have seen immense growth in the number of available input samples with the rise of big data and data augmentation techniques that have largely alleviated the mismatch between training- and test-time distributions (28). Covariate shifts remained a problem though, albeit in a different form: when training a neural network, every parameter update causes an internal distribution shift.” So they consider internal covariate shift to be a problem, but then on page 2 they again argue for alleviating covariate shift occurring due to image corruptions with BN. I find this dichotomy and mixing to be very confusing and suggest to the authors to please rewrite the sections.

Again coming back to the statement “In recent years, we have seen immense growth in the number of available input samples with the rise of big data and data augmentation techniques that have largely alleviated the mismatch between training- and test-time distributions (28)”. The authors contradict themselves on page 2 by arguing that there is a distribution shift between the train and test distributions and thus, one needs test-time adaptation to mitigate covariate shift. Given that the authors propose a technique to mitigate covariate shift, it is very strange to read in the second paragraph of the paper that this problem has already been solved.

“The Wasserstein distance provides such a method”. I do not understand this sentence. The Wasserstein distance is a metric and not a method.

Figure 2: Why does the performance not increase with a higher sample size for the distribution correction method? Intuitively, distribution matching should become easier with more samples.


### Comments to Related Work:

The references for the Data Augmentation part are not chosen well. Citing ImageNet-A for the following statement does not make sense: “The application of data augmentation, however, is somewhat limited as improving robustness against one corruption type can decrease robustness against others.” One could cite “A Fourier Perspective on Model Robustness in Computer Vision” by Yin et al. here.

The references for the statement “In the simplest case, training samples are augmented – by the application of affine transfomations or expected noise types – and explicitly included into the training set” also do not make sense in my opinion. Here, one could cite: [1] “Generalisation in humans and deep neural networks” by Geirhos et al., [2] “A simple way to make neural networks robust against diverse image corruptions” by Rusak et al., and [3] “On Interaction Between Augmentations and Corruptions in Natural Corruption Robustness” by Mintun et al.

On the references for “Adapting normalization parameters”: first, [40] was published before [5] and thus should be mentioned first. One can then write that ref [5] is concurrent work to [40], although this is a stretch since [40] has been published over a year before [5]. The current formulation is misleading. Maybe rather write something like "[5] provides further supporting evidence for XY". Further, the papers “Evaluating prediction-time batch normalization for robustness under covariate shift” by Nado et al., and “Tent: Fully Test-time Adaptation by Entropy Minimization” by Wang et al., are missing here. The TENT paper also showed BN adaptation results. Another important reference is “MEMO: Test Time Robustness via Adaptation and Augmentation” by Zhang et al which the authors should compare to since MEMO uses single sample adaptation.

### Comments on section 3:

“We assume that the prior, likelihood, and posterior come from an exponential family using a Gibbs measure so that…” please provide citations for the Gibbs measure.

“Typically, the activation distributions in CNNs cannot be represented well by parametric distributions.” Please provide citations for this statement.

“The Wasserstein distance proves to be particularly well-suited for a novel correction method”. I do not understand this sentence. The Wasserstein distance between what? The Wasserstein distance is not a method but a metric.

“Although we would ideally have access to the distributions of the clean image, this is not possible in practice. Instead we need to construct surrogate target distributions.” It is customary to refer to the distribution of clean examples as the source distribution. I am confused what the authors mean under “target distribution” here.

In eq. 5, m is not explained. It is later mentioned that M refers to the number of training distributions. What is this? How are they sampled? How large are they?

In section 3, I did not understand when the corrupted samples come into play. In the correction algorithm, one refers to a_i^m which are training samples, but the whole purpose of the correction algorithm would be to shift the test distribution closer to the training distribution. I did not understand at which point the test distribution was used.

Reading through the results, I believe the authors calculate the target distribution in eq. 8 using clean samples and sample the distribution p(a’) in eq. 7 from noisy samples. They then seek to minimize the Wasserstein distance between the two. However, this is not evident from the text at all, and I am not sure whether I understood the method correctly. This must be made much more clear.

At test time, when corrupted input samples are fed into the model, do the target vectors t need to be stored?

### Comments on section 4:

“First, we verify if our proposed surrogate distributions are expressive enough to decrease the Wasserstein distance to the clean activation distributions.” Wasserstein distance between what and the clean activation distributions?

I do not see qualitative differences between “corrected” and “uncorrected” in Fig. 5, neither in the example images nor in the histograms. I do not see that the correction layer “successfully increases the similarity between sample and target”. I also do not agree that “the correction layer introduces richer features” and find this statement unsupported.

MNIST: “We performed a coarse hyperparameter search fixing the reconstruction weight at L2 = 0.8 for all experiments. For BN we found L1 = 1.2 optimal, while for GN L1 = 1.0 provided the best results.” How were these hyperparameters selected? Were they selected on the test set? Same question for CIFAR10 and ImageNet. If these hyperparameters were selected on the test set, the results are meaningless and are not expected to generalize.

Considering the MNIST results, I do not find the improvements to be substantial. They are below 1 percent point for both BN and GN. On CIFAR, the correction method provides marginal benefits, and the CIFAR-C numbers are quite bad overall. The currently SotA CIFAR-C numbers are around 93% without adaptation: https://robustbench.github.io/#div_cifar10_corruptions_heading. I argue that showing improvements on the order of below 1 pp for subpar networks is meaningless for the robustness community since it is not clear whether this method would generalize to better models. Given how small the improvements are, I do not expect major gains for the stronger models.

ImageNet results: “This indicates that L1 correlates with the complexity resulting from the image size of the used dataset.” Unsupported statement.

It is not clear whether the results in Table 3 have been done with single sample adaptation. On ImageNet, the improvements are a bit better than on the small scale datasets. But not knowing what the concrete baselines are, judging the results is difficult. Also, a comparison to GN would be helpful on ImageNet. The baseline numbers for a ResNet50 differ from what is reported in the literature, e.g. in ref [40]. Please check the numbers to eliminate errors.

Additionally, calculating the correction terms requires additional time and resources. While the authors do not explicitly calculate gradients and update the model weights, the additional computation requirements are substantial and I do not find that the benefits outweigh the costs, especially on ImageNet.

The authors have not verified whether their correction method does not decrease clean accuracy.

---

> ### Author Response · Authors · 2022-05-04
> **Response 1**
>
> Thank you for your review!
>
> General remarks:
>
> We agree with the reviewer's sentiment that the introduction needs to be improved. We will perform a major revision of the introduction and the related work section to improve readability, clarify the background, and better motivate our work. Moreover, we will improve Section 3 to clarify the details of our proposed approach.
> Additionally, we will perform additional benchmarks on a selected set of more realistic datasets that were suggested in the reviews, i.e CIFAR-10.1, ImageNet-V2, ImageNet-R, Cityscapes.
> After carefully reviewing the related works introduced by the reviewers [2][3][4], and comparing them to [1], we will also adjust our experimental setup on the ImageNet-C and CIFAR-10-C datasets, since these works use a 15 test + 4 holdout corruption setting, instead of a test/validation splitting for the 19 individual corruptions. Also, we will perform the CIFAR evaluations using a ResNet-26. These measures should improve comparability to other methods.
>
>
> Our apologies for the format of the .pdf, the pdf-splitter we used seems to have embedded pages containing figures in some sort of image format.
> Concerning your requested changes and comments:
>
> 1 Introduction):
> - This was probably formulated slightly misleading, with "early on" we mean that this issue is well known and persists until today.
>
> - Concerning the second issue, we do not claim that larger datasets eliminate the distribution mismatch, but they certainly reduce it. As you are referring to [5], on page 3 section Distribution Gap, they argue that assessing distribution mismatch in high dimensions is difficult. Since our method is explicitly minimizing this distribution mismatch in high dimensions, it would be a great tool to investigate this issue and determine the influence of the distribution gap on ImageNet-V2 performance. We are happy to provide these results in the revision.
>
> - Concerning BN: Here our aim was to express the original intention of BN, in case this paragraph is included in the revision of the paper, we will include the footnote in the text and reformulate the sentence.
>
> - Concerning Wasserstein distance: The Wasserstein distance is a metric that enables the implementation of our method. We are sorry if this formulation was unclear, we will adjust it.
>
> - Concerning Figure 2.: Since our method uses only the current sample for the correction it does not include any information about the mini-batch statistics. This makes it independent of the number of used samples.
>
> - Other points: As mentioned above, we will perform a major rewrite of the introduction.
>
> 2 Related Work):
>
> - We did include a reference for Tent, and also present results in Figure 2.
> - We will include “Evaluating prediction-time batch normalization for robustness under covariate shift” by Nado et al., and
>   “MEMO: Test Time Robustness via Adaptation and Augmentation” by Zhang et al in the reference.
>
> 3 Method):
>
> - Concerning citation: We will provide the required citations.
>
> - Concerning Wasserstein distance: The Wasserstein distance is well suited in 1D, as it can be computed analytically and minimized in a single step. Our method uses this property, calculating multiple 1D slices of the activation distribution.
>
> - Concerning clean distributions and naming: The activation distribution of the clean image would be the ground truth distribution for a particular image. Therefore, a correction towards this distribution would be ideal. We will review the used naming convention in the paper, with regards to related work.
>
> - m denotes an input sample (i.e. an image/activation map). $m = 1,\dots,M$ , where M is the number of samples in the dataset.
>
> - Concerning method: Yes, we agree that this can be improved. We will split this section into two distinct sections (i) distribution collection and (ii) distribution correction. The target vectors are acquired on the training set and need to be stored for testing.
>
> 4) Experiments):
>
> - Concerning baseline results and generalization: As mentioned in the general remarks, we will adjust our evaluation for improved comparability. However, it has to be noted that the difference in implementation and parameters between Pytorch and Tensorflow might introduce small variations within the results.
> Generally, our results show robust behavior regarding the choice of hyperparameters. Here, Figure 7 shows the robustness with respect to the choice of hyperparameters and model initialization, as many parameter combinations lead to similar results when averaged over all severities (dashed lines). Our evaluations on different architectures also show robust behavior, as the parameters $\lambda_1$ and $\lambda_2$ were only optimized for the ResNet-50 architecture. This means that our results for the other architectures e.g. EfficientNet-B0, which showed the biggest improvement, were achieved without hyperparameter tuning.

---

> > ### Author Response · Authors · 2022-05-04
> > **Response 2**
> >
> > -Concerning benchmark(https://robustbench.github.io/#div_cifar10_corruptions_heading): If we compare the results from the best performing paper [6], using their ResNet-18 architecture, which is comparable to our used ResNet-20, we see that for the "Clean" variant without data augmentation they achieve 76.7 % (Table 2 Appendix) accuracy on CIFAR-10-C, compared to 75.95 $\pm$ 0.62 for our method. So considering similar models there is no substantial difference. The "robust accuracy" results of 92.75 \% you mentioned were achieved with additional AugMix data augmentation and for a different model architecture (WideResNet-18-2) and are therefore not comparable to our results. Therefore, as suggested in the general remarks, we will perform the CIFAR evaluations on the ResNet-26 architecture in line with other works.
> >
> > - Concerning clean data accuracy: We will include test set accuracies for uncorrupted variants of all used datasets for the revised version of the paper. Our experiments showed that there is an inherent trade-off between optimizing the hyperparameters for clean data accuracy and optimizing for corruption robustness. Therefore, all unsupervised domain adaptation methods are likely to slightly degrade on the clean dataset.
> >
> > - Concerning efficiency: We are aware of this current limitation of the method, as shown in Table 4., however, computational arguments are usually short-lived in the area of machine learning and more efficient variants of this method can be expected.
> >
> >
> > References:
> >
> > [1] Wang, D., Shelhamer, E., Liu, S., Olshausen, B. and Darrell, T., 2020. Tent: Fully test-time adaptation by entropy minimization. arXiv preprint arXiv:2006.10726.
> > https://arxiv.org/pdf/2006.10726.pdf
> >
> > [2] Khurana, A., Paul, S., Rai, P., Biswas, S. and Aggarwal, G., 2021. SITA: Single Image Test-time Adaptation. arXiv preprint arXiv:2112.02355.
> > https://arxiv.org/pdf/2112.02355.pdf
> >
> > [3] Zhang, M., Levine, S. and Finn, C., 2021. MEMO: Test Time Robustness via Adaptation and Augmentation. arXiv preprint arXiv:2110.09506.
> > https://arxiv.org/pdf/2110.09506v2.pdf
> >
> > [4] Sun, Y., Wang, X., Liu, Z., Miller, J., Efros, A. and Hardt, M., 2020, November. Test-time training with self-supervision for generalization under distribution shifts. In International Conference on Machine Learning (pp. 9229-9248). PMLR.
> > https://arxiv.org/abs/1909.13231
> >
> > [5] Recht, B., Roelofs, R., Schmidt, L. and Shankar, V., 2019, May. Do Imagenet classifiers generalize to Imagenet?. In International Conference on Machine Learning (pp. 5389-5400). PMLR.
> > http://proceedings.mlr.press/v97/recht19a/recht19a.pdf
> >
> > [6] Diffenderfer, J., Bartoldson, B., Chaganti, S., Zhang, J. and Kailkhura, B., 2021. A winning hand: Compressing deep networks can improve out-of-distribution robustness. Advances in Neural Information Processing Systems, 34.
> > https://arxiv.org/pdf/2106.09129.pdf

---

> > > ### Comment · Reviewer_bMwH · 2022-05-10
> > > **Response to the authors considering SotA CIFAR10 results**
> > >
> > > I would like to thank the authors for addressing my concerns. I have a comment considering the models they study for CIFAR10. Test-time adaptation makes sense if one can gain additional performance given some model using unlabeled test data. However, if the AugMix model performs better without test-time adaptation compared to a standard model with adaptation, a practitioner would simply use the AugMix model in eval() mode and not use test-time adaptation. For this reason, it has become customary to also try adapting the strongest models trained with e.g., additional data or augmentations to get higher performance. I would argue that both evaluations are important: It is important to study baseline models trained on the clean CIFAR10 dataset, but also look at robustified models to test whether further improvements are possible. If the second step is not done, people would simply disregard the technique and just use the better model. AugMix generalizes to other distributions as well, e.g. it performs really well on ImageNet-R; ImageNet-R constitutes a distribution shift that is quite different from ImageNet-C. I believe there should be no fundamental reason why this method could not be performed on a robustified model, and I tried to make an argument why the authors should evaluate it.
> > >
> > > The same argument also applies to MNIST and ImageNet results. Researchers would especially be interested in ImageNet results, and there, I would take the most robust ResNet50 the authors can find and adapt it with their method.

---

> > > > ### Author Response · Authors · 2022-05-10
> > > > **Response to AugMix**
> > > >
> > > > Thank you for clarifying your point! We agree with your comment concerning the evaluation of our proposed method for robust models. Including these evaluations will strengthen the paper. Therefore, for the revision, we will perform experiments using models trained with AugMix to indicate the potential improvements for robustified models.

---

### Review · Reviewer_7DYb · 2022-04-25

**Summary Of Contributions:**

This paper proposes a test-time adaptation method to improve the robustness of a given pretrained model to distribution shifts. The underlying motivation behind their method is to minimize the Wasserstein distance between representations induced by clean and corrupted samples. The authors study their approach empirically using common corruptions datasets for MNIST, CIFAR and ImageNet.

**Broader Impact Concerns:**

I am satisfied with the discussion of ethical concerns in the paper.

**Requested Changes:**

Aside from improving the writing, and incorporating additional benchmarks as suggested above, I have the following comments/questions:

1. The authors should reframe the narrative around internal covariate shift in the introduction, as follow-up works have shown that this phenomenon might not be as essential to the success of normalization methods as first believed. While the authors do caveat this in a footnote, it seems somewhat disingenuous to motivate this paper based on a hypothesis that has not really been proved.

2. Clarify how exactly the parameter estimation with a single sample works, both for the proposed method and the baseline. In BN-adapt also applied to all layers?

3. In Figure 2:
(i) Clarify the difference between the randomized/non-randomized noise type.
(ii) Why is the proposed method constant w.r.t. N? Presumably, adding more examples should help their method too. The authors should characterize the variation in the performance of the proposed method and baselines as a function of N.
(iii) Why is BN-adapt (purple) decreasing from N=1 to N=2?

4. In Figure 3:
(i) How are the bars calculated?
(ii) The bars are hard to see, especially for layers>8 given the differences in scale. The authors should replot these results with normalization (e.g., relative improvement) to make it more readable.
(iii) For lower layers (closer to the output), the difference before and after correction seems to be slight. If the proposed method was really effective, one would expect some kind of cumulative gain as we move through the network.

5. In Figure 4:
(i) What is the clean accuracy of the model?
(ii) What happens if the proposed normalization is applied to only the last-k layers?

**Strengths And Weaknesses:**

The problem of test-time adaptation of pretrained models is undoubtedly an important and interesting one---especially given the increasing prevalence of such models in both vision and language. One of the key strengths of this method in my view is that it is unsupervised and can be applied using as little as one test sample.

That being said, I do think that this paper has significant room for improvement---both in terms of writing, as well as empirical evaluation (see "Requested changes" field as well). In particular,

[Comparison to BN-adapt] On 2/3 datasets considered in the paper, the improvement of the proposed method over the baseline of BN-adapt is quite marginal (and often within confidence intervals). Given the significant added computational requirements of the proposed approach, these results do not make for a compelling case.

[Limited benchmarks] Even though the authors consider three datasets for their evaluation, they study the same set of distribution shifts applied to all these three datasets. My worry is that the performance of methods like BN-adapt and NP-C might be quite different for such (somewhat synthetic) corruption benchmarks, as compared to other natural distribution shifts like ObjectNet [1], ImageNet-R (arxiv:2006.16241), WILDS (arxiv:2012.07421), BREEDS (arxiv:2008.04859). I would find the results of this paper more convincing if the authors demonstrated similar performance levels on such benchmarks. After all, it is important to not overfit an adaptation method to a single type of downstream datasets.

[Writing] The writing of this paper can be improved, particularly in terms of clarity.
- For instance, many of the figure captions are not sufficient to parse the results in the paper (elaborated on more in the next field).
- Sections 3.1 and 3.2 are hard to follow: the authors should state their notation (what various) subscripts and superscripts mean a priori. - Several assumptions are made by the authors in the conceptual formation without justification: why an exponential family is a good characterization of the prior/posterior distributions, why r=1 is chosen for the Wasserstein distance.
- Further, it is not clear whether the authors use a single sample in all their experimental results---both for their method and the baseline of BN-adapt. In the case where the test samples could come from a mixture of shifted distributions (e.g., multiple corruptions in ImageNet-C), why do the authors expect that a single sample should be enough. To perform a non-trivial correction for each shifted distribution, I would expect that you would at least need to see 1 sample from each.


References:
[1] https://objectnet.dev/objectnet-a-large-scale-bias-controlled-dataset-for-pushing-the-limits-of-object-recognition-models.pdf

---

> ### Author Response · Authors · 2022-05-04
> **Response**
>
> Thank you for your review!
>
> General remarks:
>
> We agree with the reviewer's sentiment that the introduction needs to be improved. We will perform a major revision of the introduction and the related work section to improve readability, clarify the background, and better motivate our work. Moreover, we will improve Section 3 to clarify the details of our proposed approach.
> Additionally, we will perform additional benchmarks on a selected set of more realistic datasets that were suggested in the reviews, i.e CIFAR-10.1, ImageNet-V2, ImageNet-R, Cityscapes.
> After carefully reviewing the related works introduced by the reviewers [2][3][4], and comparing them to [1], we will also adjust our experimental setup on the ImageNet-C and CIFAR-10-C datasets, since these works use a 15 test + 4 holdout corruption setting, instead of a test/validation splitting for the 19 individual corruptions. Also, we will perform the CIFAR evaluations using a ResNet-26. These measures should improve comparability to other methods.
>
> Concerning your requested changes:
>
> 1) As mentioned above we will do a major rewrite of the introduction.
>
> 2) Yes, BN-adapt is applied to all BN layers. We will clarify the implementation details of our method.
>
> 3) While the standard corrupted MNIST, CIFAR-10, ImageNet dataset are structured into subsets of specific noise types, we mix those noise types in a random fashion. This means that for the randomized data, each mini-batch includes various different corruption types. In that regard, Figure 2. shows, that semi-supervised methods such as TENT benefit greatly from analyzing multiple images including the same type of corruption, whereas they struggle if the noise types in the batch are randomized. This behavior is not captured if the methods are only evaluated on the not randomized datasets.
> Our method and BN-adapt for the case of N=1 only act on a single test sample. This means that they are inherently independent of the number of used samples, as the statistical corrections do not use multi-sample data. In the case of Number of samples>1 BN-adapt uses multiple images to calculate the correction. Here, we again see that for the noise-specific dataset this generally improves performance, and for the randomized noise dataset, it degrades the performance. In essence, this means that for random noise scenarios, where the corruption type changes from image to image, it would be beneficial to use single sample BN-adapt.
>
> 4) The bars are calculated by first recording the activations maps of the model, using the clean CIFAR-10 images. Then we do the same thing, adding Gaussian noise to the images. The average Wasserstein distance over the dataset, for each layer, is then calculated between the activation maps of the clean, and the noisy dataset. We agree that a relative plot might be beneficial to provide better insight, especially for layers >8, we will review this option.
> The cumulative improvement is shown in Figure 4. While the correction has limited influence for certain intermediate layers, the overall accuracy improves by adding the correction to more layers.
>
> 5) Including the clean dataset accuracy directly in the plot would probably worsen the readability of the plot. We will instead include this in the figure caption and add the baseline accuracy (no corrections) to the plot. Our experiments showed, that adding the correction is most effective in the layers close to the input. Therefore, adding layers in reverse (output to input) results in very little improvement for most correction layers. We will include this graph in the plot.
>
> Other: With regards to Wasserstein distance: r=1 can be computed efficiently and minimized analytically by sorting the activations, in the case of a 1D distribution. Since we use 1D slices of our activation distribution we can exploit this property.
> Our method relies on image-wise activation statistics of the test-sample, and accumulated training set statistics. Therefore, a single sample is sufficient.
>
> [1] Wang, D., Shelhamer, E., Liu, S., Olshausen, B. and Darrell, T., 2020. Tent: Fully test-time adaptation by entropy minimization. arXiv preprint arXiv:2006.10726.
> https://arxiv.org/pdf/2006.10726.pdf
>
> [2] Khurana, A., Paul, S., Rai, P., Biswas, S. and Aggarwal, G., 2021. SITA: Single Image Test-time Adaptation. arXiv preprint arXiv:2112.02355.
> https://arxiv.org/pdf/2112.02355.pdf
>
> [3] Zhang, M., Levine, S. and Finn, C., 2021. MEMO: Test Time Robustness via Adaptation and Augmentation. arXiv preprint arXiv:2110.09506.
> https://arxiv.org/pdf/2110.09506v2.pdf
>
> [4] Sun, Y., Wang, X., Liu, Z., Miller, J., Efros, A. and Hardt, M., 2020, November. Test-time training with self-supervision for generalization under distribution shifts. In International Conference on Machine Learning (pp. 9229-9248). PMLR.
> https://arxiv.org/abs/1909.13231

---

> > ### Comment · Reviewer_7DYb · 2022-05-23
> > **Post-rebuttal update**
> >
> > I thank the authors for their response. While the authors showed willingness to address some of my major comments (writing and evaluation on additional datasets), I do agree with the other reviewers that this would constitute a major revision that would probably require to be reviewed again. I would thus advise the authors to make these changes and resubmit.

---

### Review · Reviewer_F8sr · 2022-04-25

**Summary Of Contributions:**

The authors propose a non-parametric correction layer for neural networks that reduces the distribution mismatch between training and test-time activation distributions in order to improve the robustness of these networks to covariate shift.  Experiments show improvements in classification accuracy on MNIST-C, CIFAR-10-C, and ImageNet-C.

**Broader Impact Concerns:**

The authors include a limitations and ethical concern section which I appreciate.

**Requested Changes:**

Recommendations for strengthening the work:
1. Report the original test set accuracy of each method.
2. Compare to semi-supervised test-time training algorithms in the setting where full test dataset is available.
3. Evaluate on the additional realistic/natural distribution shifts considered in prior work https://arxiv.org/abs/1909.13231
4. Improve framing/writing of intro, include more references for each claim in intro


**Strengths And Weaknesses:**

Strengths
- Unlike other semi-supervised test-time training techniques recently proposed which fine-tune or re-train the model on a pseudo-labeled test set, the method requires only a single sample at test time.

Weaknesses
- Experimental results do not report the original test set accuracy of each method, so it's hard to judge the quality of the model or see if improvements to corrupted dataset also mean improvements for the original dataset.
- Though I appreciate the fact that the method works on only a single test example, I still think it's important to benchmark against the semi-supervised test-time training algorithms in the setting where both methods have access to a large amount of test data. This would help practitioners understand which method is superior for this particular setting.
- The authors focus on covariate shifts causes by noise and corrruptions i.e. synthetic distribution shifts but do not evaluate on more realistic/natural distribution shifts such as CIFAR-10.1, ImageNet-V2, ImageNetVid-Robust. Many methods have shown to work on synthetic distribution shifts but fail to provide improvement on the realistic/natural distribution shifts (see https://arxiv.org/abs/2007.00644). Moreover, related work such as the Test-time training paper https://arxiv.org/abs/1909.13231 benchmark on both types of shifts.
- Framing of the intro is sparse on references (e.g. sentence "vulnerability of NNs to covariate shifts has been a
major concern in the past" does not cite any prior work)
- The claim in the intro that we "have largely alleviated the mismatch between training- and test-time distributions" with big-data and data augmentation ignores recent work pointing out that neural networks suffer from performance drops at test time. The argument that "covariate shift moved from test-time to training-time" doesn't make sense -- covariate shift is still a problem experienced at test time by modern neural networks.

---

> ### Author Response · Authors · 2022-05-04
> **Response**
>
> Thank you for your review!
>
> General remarks:
>
> We agree with the reviewer's sentiment that the introduction needs to be improved. We will perform a major revision of the introduction and the related work section to improve readability, clarify the background, and better motivate our work. Moreover, we will improve Section 3 to clarify the details of our proposed approach.
> Additionally, we will perform additional benchmarks on a selected set of more realistic datasets that were suggested in the reviews, i.e CIFAR-10.1, ImageNet-V2, ImageNet-R, Cityscapes.
> After carefully reviewing the related works introduced by the reviewers [2][3][4], and comparing them to [1], we will also adjust our experimental setup on the ImageNet-C and CIFAR-10-C datasets, since these works use a 15 test + 4 holdout corruption setting, instead of a test/validation splitting for the 19 individual corruptions. Also, we will perform the CIFAR evaluations using a ResNet-26. These measures should improve comparability to other methods.
>
> Concerning your requested changes:
>
> 1) We will include test set accuracies for uncorrupted variants of the used datasets (MNIST, CIFAR-10, ImageNet-C, ...) for the revised version of the paper. Our experiments showed that there is an inherent trade-off between optimizing the hyperparameters for clean data accuracy and optimizing for corruption robustness. Therefore, all unsupervised domain adaptation methods slightly degrade on the clean dataset.
>
> 2) As outlined in Figure 2., semi-supervised methods such as TENT struggle in the specific scenario of single-sample adaptation. Using multiple samples of the same noise type, however, gives these methods an inherent advantage. As a compromise solution, we suggest providing an evaluation of selected semi-supervised methods with respect to randomized corruption type datasets, where each mini-batch includes various different corruption types, similar to the MNIST evaluation in Figure 2.
>
> 3) Thank you for pointing this out. Indeed the investigated datasets include only artificial noise types; therefore, an evaluation of more realistic images would definitely provide better insights into the capabilities of the method and strengthen the paper. As mentioned above, additional results will be included.
>
> 4) As mentioned in the general remarks, we will do a major rewrite of the introduction and include all the relevant references.
>
>
> [1] Wang, D., Shelhamer, E., Liu, S., Olshausen, B. and Darrell, T., 2020. Tent: Fully test-time adaptation by entropy minimization. arXiv preprint arXiv:2006.10726.
> https://arxiv.org/pdf/2006.10726.pdf
>
> [2] Khurana, A., Paul, S., Rai, P., Biswas, S. and Aggarwal, G., 2021. SITA: Single Image Test-time Adaptation. arXiv preprint arXiv:2112.02355.
> https://arxiv.org/pdf/2112.02355.pdf
>
> [3] Zhang, M., Levine, S. and Finn, C., 2021. MEMO: Test Time Robustness via Adaptation and Augmentation. arXiv preprint arXiv:2110.09506.
> https://arxiv.org/pdf/2110.09506v2.pdf
>
> [4] Sun, Y., Wang, X., Liu, Z., Miller, J., Efros, A. and Hardt, M., 2020, November. Test-time training with self-supervision for generalization under distribution shifts. In International Conference on Machine Learning (pp. 9229-9248). PMLR.
> https://arxiv.org/abs/1909.13231

---

### Author Response · Authors · 2022-05-04
**Acknowledgements**

First of all, we would like to thank all the reviewers for the quick and detailed reviews of our paper, and the action editor for the supervision of the review process. Detailed comments are presented in the responses to the individual reviews.

---

### Decision · Action_Editors · 2022-05-30

**Recommendation:** Reject

**Comment:**

The submission introduces a non-parametric correction layer to reduce the mismatch between training and deployment activation distributions and improve a model's robustness to input distribution shifts. Empirical results are provided for MNIST-C, CIFAR-10-C, and ImageNet-C and the proposed approach is compared against Schneide & Rusak et al. (2020)'s BN+adapt approach.

Reviewers feel that the problem setting is important and relevant, and feel positive about the approach's low data requirements at test time. However, they find the empirical validation unconvincing and feel that the submission's framing of the proposed approach and treatment of related work needs improvement. The authors recognized this in their response and promised to address these concerns in a revised version of the submission.

Given that the changes required are quite substantial, it would be preferable to resubmit. I therefore recommend rejection.